# Direct RNA sequencing coupled with adaptive sampling enriches RNAs of interest in the transcriptome

Jiaxu Wang [1,4], Lin Yang[2,4], Anthony Cheng[1,4], Cheng-Yong Tham[2], Wenting Tan[1], Jefferson Darmawan[1], Paola Florez de Sessions [2] ✉ & Yue Wan [1,3] ✉

Abundant cellular transcripts occupy most of the sequencing reads in the transcriptome, making it challenging to assay for low-abundant transcripts. Here, we utilize the adaptive sampling function of Oxford Nanopore sequencing to selectively deplete and enrich RNAs of interest without biochemical manipulation before sequencing. Adaptive sampling performed on a pool of in vitro transcribed RNAs resulted in a net increase of 22-30% in the proportion of transcripts of interest in the population. Enriching and depleting different proportions of the *Candida albicans* transcriptome also resulted in a 11-13.5% increase in the number of reads on target transcripts, with longer and more abundant transcripts being more efficiently depleted. Depleting all currently annotated *Candida albicans* transcripts did not result in an absolute enrichment of remaining transcripts, although we identified 26 previously unknown transcripts and isoforms, 17 of which are antisense to existing transcripts. Further improvements in the adaptive sampling of RNAs will allow the technology to be widely applied to study RNAs of interest in diverse transcriptomes.

The advent of RNA sequencing has enabled the discovery of many new transcripts[1,2]. However, the most abundant 100 transcripts typically take up to ~60% of the sequencing reads in different tissues[3], making it difficult to detect specific transcripts of interest such as long non-coding RNAs, and discover new transcripts that were previously undetectable. As many of these poorly expressed transcripts are of biological importance, including lncRNAs and enhancer RNAs, numerous biochemical and enzymatic strategies have been developed to enrich transcripts of interest or to deplete highly abundant genes[4,5]. Traditionally, these poorly abundant transcripts can be enriched by using experimental strategies such as CaptureSeq and using biotiny-lated antisense oligo-based methods to tile along the transcripts, or by depleting other abundant transcripts using CRISPR-based methods or RNaseH-based strategies before sequencing[6–8]. However, these

methods can be tedious and long, require a large amount of starting material, and result in varying levels of enrichment efficiencies.

In recent years, the ability to directly sequence RNAs using nanopore sequencing has enabled transcript discovery and quantification without needing to convert RNA into cDNA before sequencing[9–12]. Here, native RNA strands are continuously threaded through the nanopores across a voltage differential in the 3′ to 5′ orientation. Ensuing signal perturbations in the current are converted to bases using neural network models. This allows for comprehensive mapping of RNA species (e.g., isoforms, splice variants, fusion transcripts, 3′ polyadenylation) as well as base modifications[9,10,13–16]. Direct RNA sequencing has also successfully mapped genomes of RNA viruses[12,17,18]. However, direct RNA sequencing has a lower amount of throughput than nanopore DNA sequencing, making it difficult to

[1]Stem Cell and Regenerative Biology, Genome Institute of Singapore, A*STAR, Singapore 138672, Singapore. [2]Oxford Nanopore Technologies, Singapore 138667, Singapore. [3]Department of Biochemistry, National University of Singapore, Singapore 117596, Singapore. [4]These authors contributed equally: Jiaxu Wang, Lin Yang, Anthony Cheng. ✉e-mail: paola.florezdesessions@nanoporetech.com; wany@gis.a-star.edu.sg

assay for poorly abundant transcripts. In addition to being able to sequence RNAs through the pore, nanopore sequencing also has a "read until" function whereby one could sample different sequences until one identifies the sequence of interest. In this case, one could selectively choose to sequence or not sequence transcripts of interest in the population by sequencing a short segment of each strand in the pore, mapping this to a pre-defined list of sequences of interest and triggering voltage reversal at the level of individual pores to reject undesired reads, leaving the pore ready to accept a new strand. This simplifies the enrichment and depletion of transcripts of interest.

Due to the simplicity of adaptive sampling in nanopore sequencing, it has been used to enrich specific DNA regions of interest[19–22], and used in host depletion studies where gigabase-sized reference genomes can be depleted for metagenomics analyses[23–25]. Additionally, the number of DNA adaptive sampling studies and third-party bioinformatics tools[20,22,26–29] have matured significantly over the years, facilitating the application of adaptive sampling on DNA. However, applying adaptive sampling for direct RNA sequencing is just starting to be explored[30,31]. Due to the slower motor speeds in threading RNA through the pores in direct RNA sequencing as compared to DNA sequencing, as well as different mRNA abundances, mRNA lengths and polyA tail lengths, here we test different parameters to apply adaptive sampling to direct RNA sequencing (Fig. 1a). We show that adaptive sampling can be applied to enrich transcripts of interest in RNA populations and transcriptomes and that further developments in adaptive sampling is likely to make this process more efficient in direct RNA sequencing.

## Results

### Adaptive sampling enriches RNA of interest in a population

To test adaptive sampling on direct RNA sequencing, we first in vitro transcribed three different RNAs (18S rRNA, beta-actin, and GAPDH) and included enolase2 (ENO2) RNA as provided in Oxford Nanopore Technologies' direct RNA sequencing kit to form a pool of four RNAs (Fig. 1b and Supplementary Data 1). We then performed library preparation and direct RNA sequencing of these transcripts. The sequencing library consisted of 5.6% of ACTB, 16.3% of GAPDH, 15.6% of 18S rRNA, and 62.5% of ENO2 (Supplementary Fig. 1a). During adaptive sampling, discrete chunks of bases at the 3′ terminus of each RNA strand being sequenced are simultaneously base called and mapped to transcripts of interest. In the enrichment mode, reads that align to our transcripts of interest are tagged as "stop receiving", while reads that do not align to transcripts of interest are tagged as "unblock". Additionally, reads that end before a decision can be made or could not be definitively categorized are tagged as "no decision". The "stop receiving" and "no decision" reads are classified as accepted read pool, while the "unblock" reads are rejected from continued sequencing by ejection from the pore through current reversal. Conversely, in the depletion mode, the nanopore will reverse the current and eject a transcript if it is recognized as a transcript to be depleted.

To test the efficiency of the enrichment mode of adaptive sampling for RNA, we tried different decision times for a relatively low-abundance RNA (GAPDH) to be directly enriched. Too short of a decision time might not result in enough bases being sequenced to confidently determine the identity of a transcript, and too long of a decision time will result in too many bases being sequenced and decrease the effectiveness of adaptive sampling. As such, we tested a range of decision times of 1, 3.5, and 6.5 s to identify an optimal condition (Fig. 1c). To determine whether adaptive sampling increased the absolute number of reads from GAPDH as compared to bulk sequencing, we sequenced 50% of the pores of a flow cell using adaptive sampling and the other 50% without adaptive sampling (bulk sequencing).

We obtained a good coverage across the entire transcript for all four transcripts without adaptive sampling and sequenced 195–244 K

reads for each decision time to enrich for GAPDH transcripts, while keeping a similar number of reads sequenced in control bulk sequencing (Supplementary Fig. 1b and Supplementary Data 1). As direct RNA sequencing occurs from the 3′ end of the transcript, we started observing a 5′ end depletion of non-GAPDH transcripts (18S rRNA, ACTB and ENO2) at 1 s of adaptive sampling, and a stronger depletion at 3.5 and 6.5 s, while we sequenced full-length transcripts of GAPDH at all three time points (Fig. 1d and Supplementary Fig. 1c–f). To determine whether we obtained an increase in the number of reads on GAPDH in adaptive sampling as compared to bulk sequencing, we mapped the reads to the four RNAs during the different decision times in adaptive sampling and their respective bulk sequencing.

We did not observe a net increase in the number of bases and reads mapped to GAPDH at 1 s of adaptive sampling (Fig. 1e), suggesting that 1 s decision time could be too short to effectively enrich RNAs. At a longer decision time of 3.5 s, we obtained 16,227 reads in bulk sequencing and 19,867 reads in adaptive sampling, which is a 22% increase in read count (Fig. 1f). The median length of the rejected reads increased from 276 bases to 372 bases as the decision time increases from 1 to 3.5 s (Supplementary Fig. 1g), agreeing with the expectation that more bases are sequenced when a longer data acquisition time is used. At an even longer decision time of 6.5 s, we obtained 16,149 reads that are mapped to GAPDH in bulk sequencing and 17,252 reads in adaptive sequencing (7% increase, Fig. 1g). This indicates that at 6.5 s decision time, too many bases have been sequenced before a decision is made for enrichment to be effective. In addition to the number of reads that are mapped to GAPDH, we also calculated the total number of bases that are mapped to GAPDH in adaptive sampling versus bulk sequencing. We observed a 26.5% and 8% increase in the number of bases mapped to GAPDH at 3.5 and 6.5 s of decision time of adaptive sampling as compared to bulk sequencing (Fig. 1f, g). Additionally, we observed a low false rejection rate of 2.8–5.7% for GAPDH in the rejected pool, at different decision times, indicating that adaptive sampling is performing accurately as expected (Fig. 1h). These results converge that 3.5 s decision time is ideal to enrich transcripts of interest.

### Adaptive sampling depletes RNAs of interest in a population of RNA

As a few of the most abundant transcripts can occupy most of the sequencing reads[3], depletion of abundant transcripts can allow deeper sequencing of the less abundant transcripts in the pool. We next tested adaptive sampling in the depletion mode to deplete an abundant transcript of interest. We chose ENO2 as it is the most abundant transcript out of the 4 RNAs (62.5%). As we had previously observed that 1 s was too short (due to the translation speed of 70 bases per sec) to make confident decisions and 3.5 s was sufficient to enable efficient enrichment, we tested an intermediate range of decision times from 2–4.5 s for the depletion mode of adaptive sampling, using 50% of pores of the flow cell for bulk sequencing and 50% for adaptive sampling, and sequencing to similar read depths at each decision time (Supplementary Fig. 2a).

We started observing the depletion of ENO2 at a 2-s decision time, with only short ENO2 transcripts being sequenced while the other three transcripts are sequenced to full length (Fig. 2a and Supplementary Data 2). This results in a decrease in the number of reads mapped to ENO2 from 54,580 reads in bulk sequencing to 12,730 reads in adaptive sampling (Fig. 2b), and a net increase of 34% in the number of reads mapped to non-ENO2 transcripts in adaptive sampling (27,469 in bulk sequencing vs. 36,840 in adaptive sampling, Fig. 2b). However, as we can still see longer reads of ENO2 being present at 2 s of decision time (Fig. 2a), this suggests that 2 s of decision time is too short because ENO2 transcripts are not being depleted properly. As different reads could have different lengths, we also confirmed the increase in output in adaptive sampling at the base level (Fig. 2b–e). As expected,

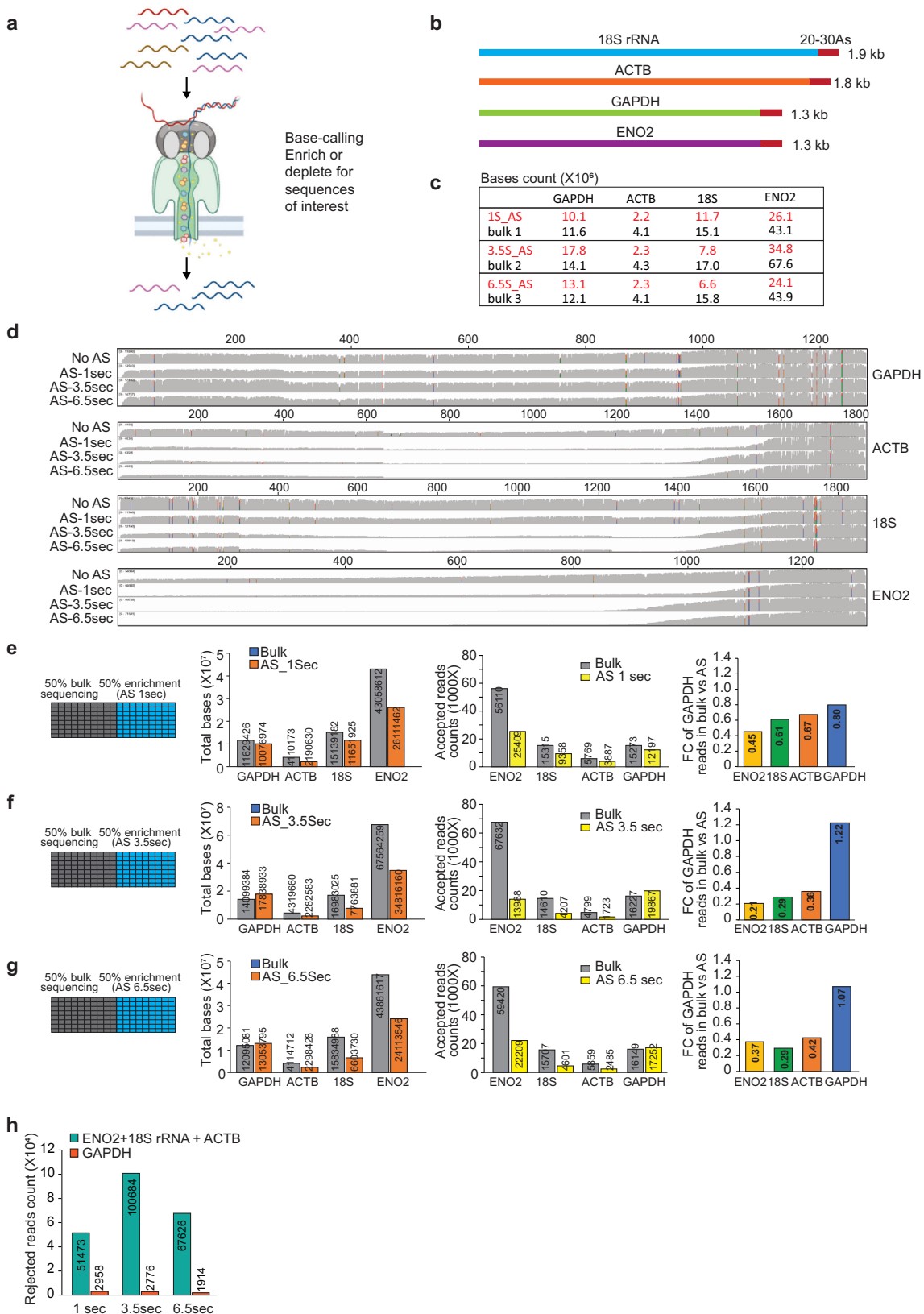

the number of bases that are sequenced before a decision is made to reject or accept a read increases from 291 to 376 bases as the decision time increases from 2 to 4.5 s (Supplementary Fig. 2b).

Like the enrichment mode, we observed that 3.5 s of decision time is sufficient for efficient depletion of ENO2. At 3.5 s of decision time, the number of reads mapped to ENO2 decreased from 56,695 reads in

bulk sequencing to 12,006 reads in adaptive sampling (4.72 fold decrease), while reads on the three other transcripts increased from 28,844 reads in bulk sequencing to 37,174 reads in adaptive sampling (29% increase). We also observed a similar decrease and increase in the number of reads mapped to ENO2 and other transcripts at 4.5 s of decision time respectively. ENO2 reads decreased from 50,139 reads in

**Fig. 1 | Adaptive sampling enriches for transcripts of interest. a** Schematic showing the nanopore adaptive sampling sequencing workflow. **b** Schematic of four transcripts that were generated by in vitro transcription (IVT), and then used for adaptive sampling sequencing. The four genes are labeled with different colors as shown. **c** Table showing the parameters for break time (decision times) and the obtained read numbers during different decision times for adaptive sampling in bulk sequencing and in adaptive sampling at different decision times. The base counts from Adaptive sequencing are labeled orange. **d** IGV plots showing the distribution of read coverage along the length of GAPDH, ACTB, 18S rRNA, and ENO2 after GAPDH enrichment by adaptive sampling. We tested different decision times (1, 3.5, or 6.5 s) for adaptive sampling. Left: Schematic showing parameters for bulk and adaptive sequencing, using 50% of the pores of a flow cell for bulk

sequencing and 50% of the pores for adaptive sequencing at 1 s (**e**), 3.5 s (**f**), and 6.5 s (**g**), decision times. Left Middle: Bar plots showing the total number of bases obtained on GAPDH, ENO2, 18S rRNA, and ACTB using adaptive sampling and bulk sequencing. Right Middle: Bar plots showing the number of reads mapped to GAPDH, ENO2, 18S rRNA, and ACTB in normal bulk sequencing (from 50% of the pores) and in adaptive sampling sequencing (from the other 50% of the pores) at different decision times. Right: Bar plots showing the percentage of accepted reads in adaptive sequencing as compared to bulk sequencing for each transcript. The color labels are as shown. **h** Bar plots showing the number of reads from GAPDH, ENO2, 18S rRNA, and ACTB in the rejected reads pool at different decision times. The color labels are as shown.

bulk sequencing to 9967 reads in adaptive sampling, while reads on other transcripts increased from 26,542 reads in bulk sequencing to 33,631 reads in adaptive sampling (27% increase, Fig. 2d, e). We also observed that 99.9% of the rejected reads belong to ENO2 at each decision time (Fig. 2f), with less than 0.1% of falsely rejected reads, suggesting that the depletion mode of adaptive sampling is more accurate than that of enrichment mode (Figs. 1h and 2f).

In addition to depleting a single RNA from the population, we tested depleting two transcripts at the same time. Depleting both ENO2 and GAPDH from the population simultaneously using 3.5 s decision time resulted in a decrease of ENO2 from 52,850 to 12,786 reads (4.1X reduction), a decrease of GAPDH from 12,113 to 4268 reads (2.84X reduction, Fig. 2g and Supplementary Fig. 2c), and a collective increase of the other two transcripts (18S rRNA and ACTB) from 15,048 to 18,952 reads (1.26X increase, Fig. 2g). At the base level, we obtained 18.03 million reads for 18S rRNA and ACTB in bulk sequencing, and 22.15 million bases in adaptive sampling (22.8% increase, Fig. 2g). Studying the population of rejected reads indicated that >99.9% of the rejected reads belong to GAPDH and ENO2, indicating that depleting two transcripts simultaneously is effective with no falsely rejected reads (Fig. 2h).

## Adaptive sampling can be applied to transcriptomes to deplete or enrich specific RNA populations

As the efficiency of enrichment/depletion in a small group of transcripts might differ from that in a transcriptome, we applied adaptive sampling to study the transcriptome of the pathogenic fungi *Candida albicans*. *Candida albicans* is a commensal fungus that is typically found on the mucosal surfaces of healthy individuals[32]. However, it can become invasive and infect individuals when they are immunocompromised. We first performed two replicates of direct RNA sequencing and detected 3877 transcripts with at least 5 reads in each replicate. We observed that the top 150 transcripts occupy 55% of all sequencing reads in the *Candida albicans* transcriptome (Fig. 3a), making it difficult to capture the less abundant transcripts.

To determine whether we can perform adaptive sampling to capture a small fraction of the *Candida albicans* transcriptome, we selected 319 transcripts in the 80th–90th quantile of gene expression (Fig. 3b and Supplementary Data 3). These 319 transcripts include 41 genes that are important for *Candida albicans* to transition from yeast to hyphae stages for infection and comprise about 5.4% of its transcriptome. Using adaptive sampling at 3.5 s decision time to enrich for these 319 transcripts, we observed that only 1.9% of all rejected reads belonged to the 319 transcripts, indicating a high accuracy in selecting for genes of interest at a transcriptome level (Fig. 3c). To determine whether adaptive sampling increased the absolute number of reads from the 319 transcripts as compared to bulk sequencing, we sequenced 50% of a flow cell using adaptive sampling and the other 50% of the same flow cell without adaptive sampling to control for any variability between library preparations, flow cells and time of the runs. From 48 h of sequencing, we obtained a total number of 15,586 and 12,480 reads for 319 transcripts in bulk sequencing and adaptive

sampling respectively, indicating that adaptive sampling does not result in more reads in the enrichment mode (Fig. 3d and Supplementary Fig. 3a).

We next tested a depletion experiment to deplete away the rest of the transcripts in the transcriptome (4997 transcripts, 95% of our transcriptome) to enrich our 319 transcripts of interest. We first checked that the depletion mode indeed works by examining the transcripts in the rejected pool. We observed that 99.7% of the reads in the rejected pool belong to the transcripts for depletion, while only 0.3% belonged to our transcripts of interest, indicating that the depletion worked as expected (Fig. 3e and Supplementary Data 4). To determine whether the depletion mode results in more reads from the 319 genes, we again sequenced 50% of a flow cell using the depletion mode of adaptive sampling and the other 50% without adaptive sampling for 48 h. We obtained a total of 35,057 reads for 319 genes in adaptive sampling, as compared to 30,912 reads in bulk sequencing (an increase of 13.4%, Fig. 3f and Supplementary Data 4), indicating that the depletion mode of adaptive sampling does increase the number of reads belonging to transcripts of interest (Supplementary Fig. 3b).

In addition to directly enriching for a specific RNA(s) of interest, another common strategy to enrich transcripts is to deplete the most abundant genes so that the rest of the transcriptome can be sequenced deeper. Following this logic, we depleted the 150 abundant transcripts in the *Candida albicans* transcriptome that comprise 55% of the sequencing reads in bulk sequencing. In total, 98.6% of the transcripts in the rejected population belong to these 150 transcripts, indicating that adaptive sampling is working (Fig. 3g and Supplementary Data 5). In total, 50–50% sequencing with and without adaptive sampling for 68 h showed an 11.15% increase in the reads belonging to the rest of the transcriptome (Fig. 3h and Supplementary Fig. 3c), demonstrating that depleting the most abundant transcripts can result in deeper sequencing of the remaining RNAs. This strategy is particularly helpful for transcript discovery when one is interested in identifying new and poorly expressed RNAs in the transcriptome.

## Longer transcripts are more efficiently depleted and enriched in the transcriptome

To determine whether there are features that determine an RNA's ability to be enriched or depleted in the transcriptome, we tested the impact of the length and abundance of the transcripts in adaptive sampling. To do this, we tested the efficiency of depletion for transcripts that are between 200–400, 400–600, 600–1000 bases, and longer than 1000 bases, in the sample where we depleted 4997 transcripts. We observed that transcripts that are shorter than 600 bases are not depleted effectively and that the longer transcripts are better depleted (Fig. 4a and Supplementary Fig. 4a). Shorter transcripts are also depleted less efficiently when we apply adaptive sampling in enrichment mode to enrich the 319 transcripts (Supplementary Fig. 4b). To test the dependency of transcript lengths with their ability to be enriched, we investigated the efficiency of enrichment in different transcripts when we deplete the top 150 abundantly expressed genes. We observed a weak trend whereby

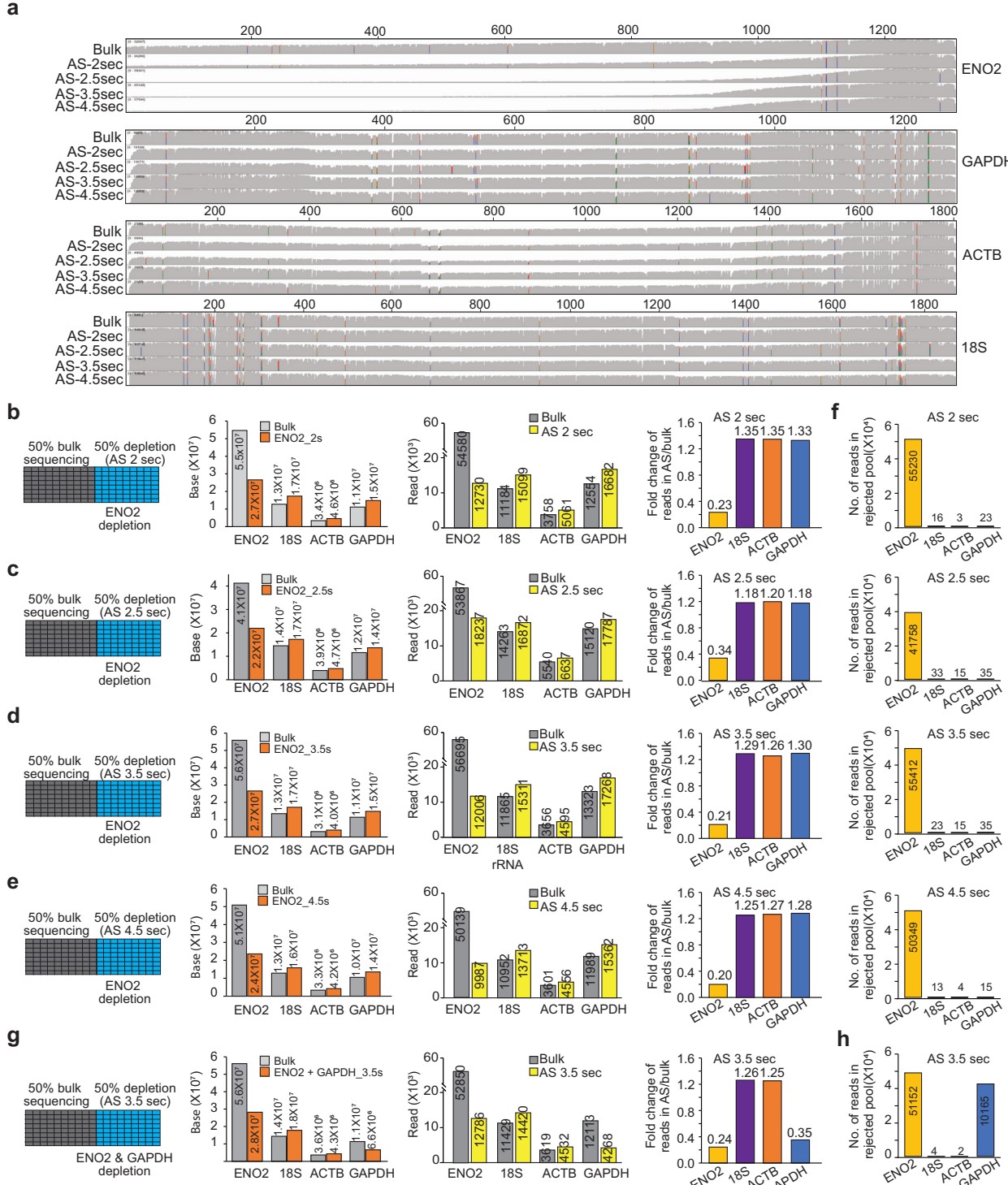

**Fig. 2 | Adaptive sampling depletes transcripts of interest. a** IGV plots showing the distribution of read density along GAPDH, ACTB, 18S rRNA, and ENO2 after ENO2 depletion. We tested different decision times (2, 2.5, 3.5, or 4.5 s) for adaptive sampling. Left: Schematic showing parameters for bulk and adaptive sequencing, using 50% of the pores of a flow cell for either bulk sequencing or adaptive sampling at 2 s (**b**), 2.5 s (**c**), 3.5 s (**d**), and 4.5 s (**e**) decision times, to deplete ENO2. Left Middle: Bar plots showing the total number of bases obtained on GAPDH, ENO2, 18S rRNA, and ACTB using adaptive sampling and bulk sequencing. Right Middle: Bar plots showing the number of reads mapped to GAPDH, ENO2, 18S rRNA, and ACTB in normal bulk sequencing and accepted pool of adaptive sampling at different decision times. Right: Barplots showing the percentage of accepted pool reads in adaptive sampling as compared to bulk sequencing for each transcript. The color labels are as shown. **f** Bar plots showing the number of reads from GAPDH, ENO2, 18S rRNA, and ACTB in rejected-reads pool at different decision times. **g** Left: Schematic showing parameters for bulk and adaptive sequencing for depleting ENO2 and GAPDH, using 50% of the pores of a flow cell for either bulk sequencing or adaptive sequencing at 3.5 s decision time, to deplete ENO2 and GAPDH. Left Middle: Bar plots showing the total number of bases obtained on GAPDH, ENO2, 18S rRNA, and ACTB using adaptive sampling and bulk sequencing at 3.5 s decision time. Right Middle: Bar plots showing the number of reads mapped to GAPDH, ENO2, 18S rRNA, and ACTB in bulk sequencing and accepted pool of adaptive sampling sequencing at 3.5 s decision time. Right: Bar plots showing the percentage of accepted pool reads in adaptive sequencing as compared to bulk sequencing for each transcript. **h** Bar plots showing the number of reads from GAPDH, ENO2, 18S rRNA, and ACTB in the rejected reads pool after depleting ENO2 and GAPDH using 3.5 s decision time. The color labels are as shown.

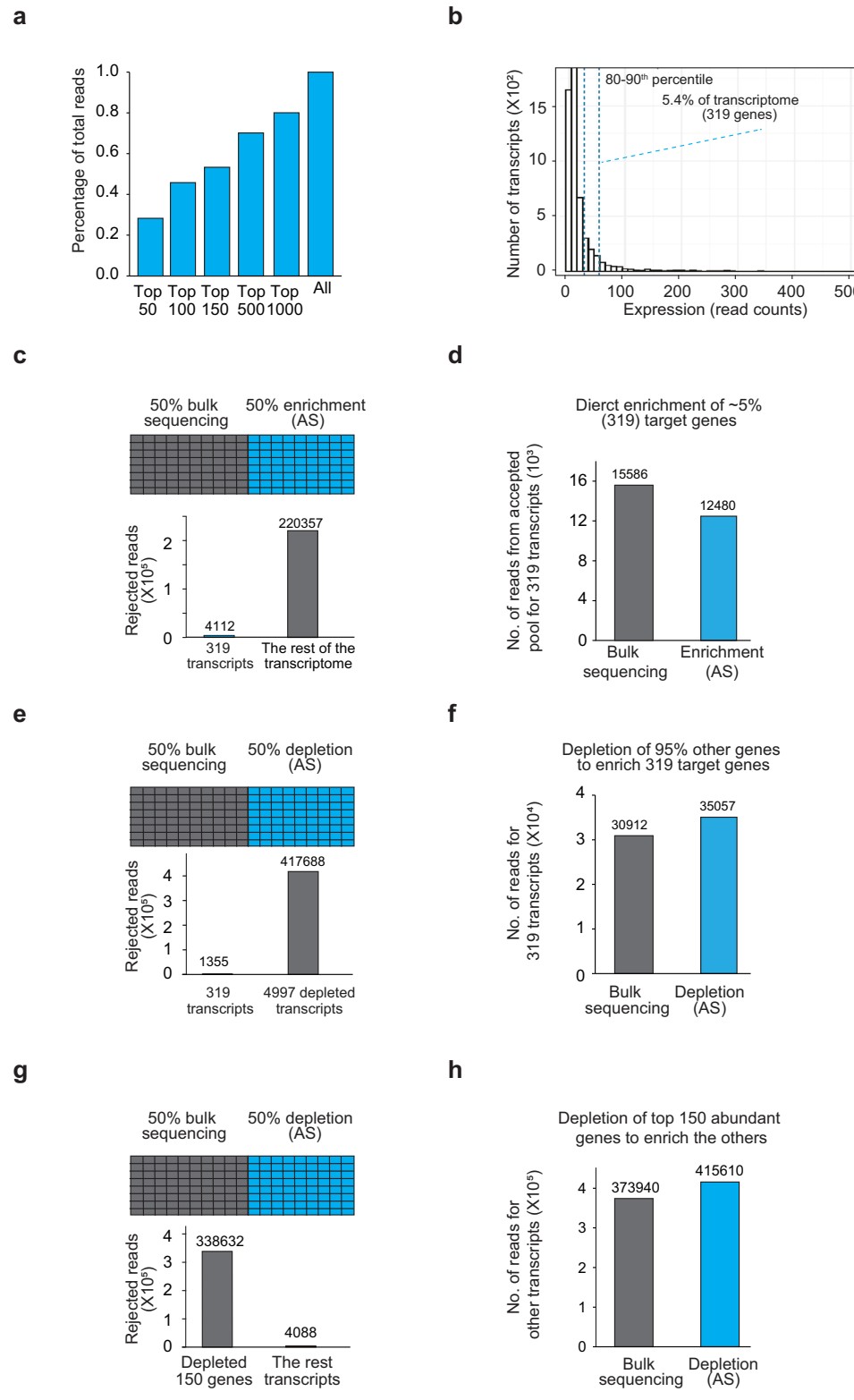

longer transcripts are better enriched (Fig. 4b and Supplementary Fig. 4c). These results agree with the 3.5 s decision time in adaptive sampling, which translates to around 350 bases being sequenced before the decision is made to accept or reject a read. In addition to length, binning transcripts according to their abundance, after the top 150 transcripts are depleted, showed that poorly abundant transcripts are less enriched as compared to more abundant transcripts (Supplementary Fig. 4d). In contrast, binning transcripts according to their polyA tail length did not show differences in their

ability to be depleted in the 4997 transcripts (Supplementary Fig. 4e, f).

As most of the sequencing reads in a cell are mapped to abundant transcripts, we investigated whether there are novel transcripts or isoforms in the *Candida albicans* transcriptome by depleting all known transcripts in its genome using existing annotations[33–36]. We identified 600 reads that fall on 26 new transcripts and variations of transcripts that have not been previously reported in existing annotations or literature (Supplementary Data 6). These transcripts are also found in

**Fig. 3 | Adaptive sampling enriches RNA populations in the transcriptome.**
**a** Bar plots showing the percentage of reads that the top 50, 100, 150, 500, and 1000 expressed genes occupy in bulk sequencing data. **b** Bar plots showing the distribution of transcript abundance in the *Candida albicans* transcriptome. In total, 319 genes from the 80–90th percentile of abundant genes are selected for enrichment. **c** Number of total reads obtained from 319 transcripts using either 50% bulk sequencing or adaptive sampling (enrichment mode). Top: Schematic showing the experimental design: 50% of the pores are set for bulk sequencing (gray) and adaptive sampling (blue) in a flow cell. Bottom, Bar plots showing the number of reads from the selected 319 genes and the rest of the transcriptome in the rejected read-pool after adaptive sampling. **d** Bar plots showing the number of reads belonging to the 319 selected genes in bulk sequencing and adaptive sampling. **e** Number of total reads obtained from 319 transcripts using either 50% bulk

sequencing or adaptive sampling after depleting 95% of the transcriptome. Top: Schematic showing the experimental design: 50% of the pores are set for bulk sequencing (gray) and adaptive sampling (blue) in a flow cell. Bottom, Bar plots showing the number of reads from the selected 319 genes and the rest of the transcriptome in the rejected read-pool after adaptive sampling. **f** Bar plots showing the number of reads belonging to 319 genes and the other 4997 genes upon depleting 95% of the transcriptome. **g** Top: Schematic showing the experimental design: 50% of the pores are selected for bulk sequencing and adaptive sampling in a flow cell. Bottom, Bar plots showing the number of reads belonging to the top 150 genes and the rest of the transcriptome in the rejected read-pool. **h** Bar plots showing the number of reads belonging to the rest of the transcriptome from 50% of pores that performed bulk sequencing and adaptive sampling by depleting the top 150 abundant genes. The color labels are as shown.

bulk sequencing and are not enriched in our adaptive sampling library (Supplementary Fig. 5a). To determine why these transcripts are poorly enriched in adaptive sampling, we plotted the distribution of their abundance and length and compared that to annotated *Candida albicans* transcripts. We observed that these novel transcripts are generally shorter in length (Supplementary Fig. 5b) and much lower in abundance (Supplementary Fig. 5c) and as compared to annotated transcripts. As length and abundance both contribute to efficiency in enrichment, these two factors probably contribute to the transcripts being poorly enriched in adaptive sampling. Out of these 26 transcripts, 17 of them are antisense to existing transcripts or are transcribed in between two existing transcripts, indicating that the *Candida albicans* transcriptome is more complex than previously appreciated (Fig. 4c, d).

Last, as adaptive sampling results in frequent current reversal to eject off-target transcripts, we tested the effect of adaptive sampling on pore health using 50–50% with and without adaptive sampling for sequencing on a single flow cell. We observed that the number of single pores, which corresponds to the number of available pores, decay at a similar rate for adaptive sampling and bulk sequencing (Supplementary Fig. 5d, e), confirming that the health of the pores is similar with and without adaptive sampling at these levels of enrichment.

## Discussion
The discovery of new and poorly abundant transcripts, as well as their gene organization, remains a challenge in transcriptomics because RNA expression varies by $10^5$ across different transcripts[37]. As such, traditional RNA sequencing needs to sequence through all the abundant transcripts before we can discover and better understand the structure and function of the poorly expressed transcripts. Here, we tested adaptive sampling using direct RNA sequencing to define the parameters that enable enrichment and depletion of transcripts. We applied adaptive sampling to individual RNAs and populations of RNAs in an in vitro transcribed pool and the transcriptome, respectively. We showed that depleting unwanted transcripts can enrich RNAs of interest. Additionally, longer and more abundant transcripts are more efficiently enriched and depleted in RNA populations and we did not observe a decrease in the health of the pores involved at the enrichment levels observed in these experiments. We also identified 26 new transcripts in the *Candida albicans* transcriptome, contributing to our understanding of the complexity of its transcriptome.

One main caveat of the current version of adaptive RNA sampling strategy is that it takes a relatively long amount of time to decide whether to reject or accept a read (3.5 s plus the amount of time to base call, map and decide) as compared to the average length of an mRNA. As the average length of a eukaryotic RNA ranges from 1200–1800 bases[38], this decision time translates to ~350 bases, and currently limits the effectiveness of adaptive RNA sequencing on the transcriptome. As such, although we do observe significant enrichments of transcripts of interest in the accepted pool, the total number

of reads belonging to the transcripts of interest only increased by 30% in the IVT pool and 13% in the depletion mode of adaptive sampling for the *Candida albicans* transcriptome. We believe that adaptive sampling of RNA will be more effective in a population of long RNAs, whereby the number of bases that need to be sequenced to determine the read's identity becomes a smaller fraction of its total length. Additionally, reducing the amount of computational time needed to determine the identity of the transcript, to eject the read of interest, and to receive new RNA molecules will further improve adaptive sampling in future. As this is an early rendition of adaptive sampling on RNA and an area of active research, we believe that future improvements in processing and pore longevity will also further facilitate the utility and adoption of adaptive sampling in transcriptomes.

## Methods
### Generation of in vitro transcription (IVT) RNA and polyA RNA
The RNA benchmarkers were generated by HiScribe TM T7 High Yield RNA Synthesis Kit (NEB #E2040S) from PCR products. *Candida albicans* transcriptome total RNA was extracted by TRIzol, and the polyA RNA was purified by Poly(A)Purist™ MAG Kit (Thermo Fisher, AM1922).

### Library preparation
In total, 200 ng total of in vitro transcribed (IVT) RNAs mixed in equimolar ratios or 800 ng poly-A enriched *Candida albicans* transcriptome was used for library preparation with the direct RNA sequencing kit (SQK-RNA002, Oxford Nanopore Technologies). Libraries were loaded on MinION R9.4.1 flow cells and sequenced on the GridION-Mk1.

### Adaptive sampling setup using MinKNOW
Adaptive sampling runs were set up in MinKNOW GUI (version 22.10.5), with modifications to decision times made to "/opt/ont/minknow/conf/package/sequencing/sequencing_MIN106_RNA.-toml" under the "*break_time_in_seconds*" parameter. For depletion runs, an additional parameter modification was applied (deplete_stop_receiving_min_sequence_length = 600). Live basecalling was performed using Guppy 6.3.8, using the high accuracy (HAC) model. Enrichment or depletion targets were provided as FASTA indexes (".mmi"), using the minimap2 indexing preset ("sr") for short reads. Each MinION flow cell was divided in half, with 256 channels running regular bulk sequencing and the other 256 channels performing adaptive sampling with the appropriate gene panels. For IVT adaptive sampling, runs proceeded for ~3 h. For *Candida albicans* experiments, runs were stopped at either the 48-h or 68-h mark.

### Data preprocessing
FASTQ output files that passed the Q-score threshold value of 7 were used for data analysis. Output from adaptive sampling were grouped based on adaptive sampling decisions as reported in the "adaptive_sampling_[flow cell id]_[run id].csv" file generated during the run, with "*stop_receiving*" or "*no_decision*" reads included in the accepted pool

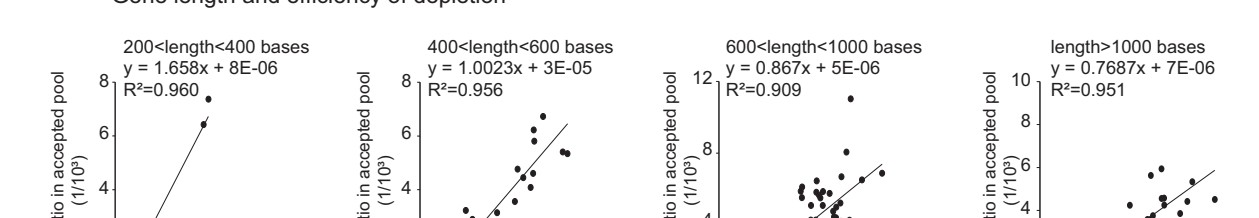

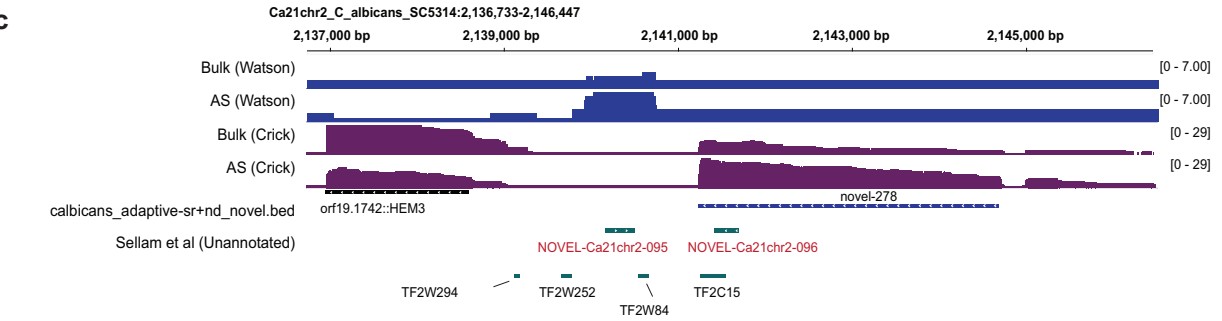

**Fig. 4 | Longer transcripts are enriched better in the transcriptome.**
**a** Scatterplots showing the percentage of reads belonging to the unenriched 4997 transcripts of different lengths (200–400 bp, 400–600 bp, 600–1000 bp, >1000 bp) in bulk sequencing (X-axis) and in adaptive sampling (Y-axis), after enrichment for 319 genes. **b** Scatterplots showing the percentage of reads belonging to the rest of the transcriptome with different lengths (200–400 bp,

400–600 bp, 600–1000 bp, >1000 bp) in bulk sequencing (X-axis) and in adaptive sampling (Y-axis), after depleting the top 150 genes. **c**, **d** IGV plots showing the location and read count of newly identified transcripts through adaptive sampling by depleting all known transcripts. The Waston strands are labeled blue, and Crick strands are labeled brown.

and "*unblock*" reads included in the rejected pool. In the split flow cell setup for *Candida albicans* experiments, reads were first assigned to regular bulk sequencing or adaptive sampling channels utilizing read_id and channel information provided in the "sequencing_summary_[flow cell id]_[run id].txt" file. Reads in the respective categories were aligned to a reference consisting of sequences of IVT transcripts or a reference transcriptome for *Candida albicans* using minimap2 (version 2.24) under the option "-ax map-ont"[39]. Aligned reads were subsequently filtered using SAMtools (version 1.14) with the -F 2308 flag to remove unmapped, supplementary and secondary alignments[40].

### Read enrichment calculation

For adaptive sampling with IVT RNA mixes, the efficiency of enrichment across various decision times was assessed by calculating the proportions of each transcript within the pooled of accepted reads (combination of "*stop_receiving*" and "*no_decision*" outcomes).

For adaptive sampling experiments with *Candida albicans*, the 50:50 split flow cell setup allowed for direct comparison of read counts for transcripts of interest under regular bulk sequencing and adaptive sampling conditions, keeping consistent the library, flow cell and run time.

### Pore health analysis

Pore health analysis was performed using a *Candida albicans* 68 h run in the split flow cell configuration. Pore state information at each pore scan (occurs at 1.5 h intervals) was obtained from the "pore_scan_data_[flow cell id]_[run id].txt" file generated with each MinKNOW run. The single_pore state, corresponding to the number of active pores available for sequencing was computed for each half of the flow cell for a direct comparison of pore health and sequencing capacity during regular bulk sequencing and adaptive sampling runs.

### Reporting summary

Further information on research design is available in the Nature Portfolio Reporting Summary linked to this article.

## Data availability

The data supporting the findings of this study are available from the corresponding authors upon request. The raw sequences data was uploaded to European Nucleotide Archive https://www.ebi.ac.uk/ena/browser/home. The accession numbers are uploaded and archived at ENA with accession PRJEB70914.

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

## Acknowledgements

We thank members of the Wan lab and Oxford Nanopore for helpful discussions. Y.W. is supported by funding from A*STAR Investigatorship, National Research Foundation of Singapore, EMBO Young Investigatorship and CIFAR Azrieli global scholar fellowship. J.W. is supported by funding from A*STAR CDA grant (202D8067). We thank Dan Turner, Libby Snell, Miguel Freitas Ribeiro Gaspar Reis and Miles Benton for reading the manuscript and helpful discussions.

## Author contributions

Y.W. and P.F.d.S. conceived the project. Y.W., P.F.d.S., J.W. and L.Y. designed the experiments. J.W., L.Y., W.T. and J.D. performed the experiments. L.Y., A.C. and C.T. performed the computational analysis. Y.W., J.W. and L.Y. organized and wrote the paper with all other authors.

## Competing interests

P.F.d.S., L.Y. and C.T. are employed by Oxford Nanopore Technologies. The remaining authors declare no competing interests.
