## [Peer Review File · Nature Communications]

Direct RNA sequencing coupled with adaptive sampling enriches RNAs of interest in the transcriptomeREVIEWER COMMENTS

Reviewer #1 (Remarks to the Author):

In this paper Yang and colleagues test aspects of adaptive sampling on RNA sequencing on Oxford Nanopore sequencers. They use a mixture of synthetic and real samples to determine the relative benefits of the approach and how one might optimise the approach. The data within the paper are interesting and useful.

However, the paper suffers from a common flaw in adaptive sampling papers in that at no point is absolute enrichment calculated for the experiments presented. The majority of data are presented as proportions or some other normalised variant. This does not help the reader (see below for a detailed discussion of figure 1). In my view it is essential to present enrichment in absolute terms using the number of bases sequenced on target in adaptive sampling experiments compared with control experiments. As shown below this highlights variability in experiments that users may expect to see. Crucially, this removes the reliance on the proportion of reads in the accepted or rejected fraction which is really a measure of the efficiency of read classification - it is not a measure of the efficiency of adaptive sampling.

Figure 1 A-C is clear.

Figure 1D is also clear although it is unclear why the coverage for 18S rRNA does not drop from 3' to 5' as every other transcript does?

Figure 1E and F are somewhat confusing as the data are presented grouped by transcript (E) or grouped by time (F). Why is this the case? The data are difficult to intuitively understand. The authors propose a 2.3 fold increase in the "amount of GAPDH that is present in the accepted pool". This statement is misleading. The accepted pool is defined as those reads with a stop receiving classification or a no decision classification. If every read was correctly classified, then all GAPDH reads would be in the accepted pool. This would give a proportion of 100% reads in the accepted pool but tells us nothing about an increase in amount. In order to know this, one needs to know read numbers and numbers of bases. Read numbers are reported in supplementary table 1. This does indeed show that some absolute enrichment is seen in the number of reads that are observed in two of the experiments - in the 1 second AS time, there are 1922 accepted GAPDH reads (i.e a 1.7 fold increase). In the 3.5s AS experiment there are 2187 reads (i.e a 1.96 fold improvement. However, in the 6.5s experiment there are only 791 reads - i.e a drop in yield (a 0.71 fold "improvement").

This most likely reflects the yield of each of the experiments. Indeed - the control experiment generates a total of 9,225 reads, the 1s 20,363 (including rejected reads), the 3.5s 20,086 and the 6.5s only 7,745 reads. Thus the drop in yield is the underlying explanation for the different result. Remarkably if one calculates the relative proportions of reads mapping to each transcript in the pools and includes sequenced and rejected counts, one finds that the relative proportions of reads identified as coming from each transcript doesn't significantly change (exactly as one expects!). It is merely the length of the reads that will differ.

In my view the experiment has done exactly what it should have done, but that is hard to see in the way it is presented. The measurements of "enrichment" are not accurate for either read number or bases sequenced as presented.

The same presentation arguments apply to figure 2. Indeed, I think the same issues are present throughout the remainder of the manuscript. By measuring the fraction of reads in the accepted pool the analysis fails to fairly represent the total data.

In figure 3 panels D, H and I the absolute change in read number is relatively small. Is there a way to determine the significance of this? In the absence of measures of absolute yield it is difficult to determine if the read number measures are sufficient.

Reviewer #2 (Remarks to the Author):

The authors of "Direct RNA sequencing coupled with adaptive sampling enriches RNAs of interest in the transcriptome" demonstrate that adaptive sampling applied to direct RNA nanopore sequencing can enrich or deplete target transcripts during sequencing without the need for biochemical techniques prior to sequencing. They used both a sample of 4 different IVT transcripts of similar lengths and the fungus *Candida albicans*. They demonstrated that the adaptive sampling of RNA molecules doesn't unnecessarily degrade the nanopore and that there are significant fold enrichments of target transcripts when compared to bulk phase. However, as stated in the discussion by the authors, the time to decision is too long and at best results in only a 10% increase in number of on target reads, suggesting that another flowcell worth of sequencing would result in more on and off target transcripts sequenced than if adaptive sampling was used. There needs to be extensive improvements in the time to decision before adaptive sampling can effectively be used with direct RNA nanopore sequencing. This will be of even more importance if ONT begins using their new faster RNA motor as advertised last summer.

Below are specific comments to the authors:

The introduction is sparse on details that would prepare the reader for the experiments to follow. More details on the effectiveness of the various enrichment or depletion methods mentioned in the introduction would help put in perspective what is possible and where this method fits into the field of target transcript enrichment/depletion and DNA adaptive sampling.

The claim in the second sentence of the introduction should have one or more citations. "However, the most abundant 100 transcripts typically take up more than 50% of the sequencing reads, making it difficult to detect specific transcripts of interest such as long non coding RNAs, and to discover new transcripts that were previously undetectable."

The claim in the last sentence of the first paragraph of the introduction should have one or more citations "While effective in certain cases, these methods can suffer from poor enrichment efficiencies and degradation of RNA material during the enrichment steps, complicating the experimental process."

It is unclear from the initial testing with the 4 IVT RNAs (ACTB, GAPDH, 18s rRNA, and ENO2) if the poly(A) tail was included as part of the decision time, and how different poly(A) tail lengths may influence the method's enrichment or depletion of targets.

I'm a little surprised that the median number of nucleotides sequenced for the rejected transcripts were 276 for 1 second of sequencing. It has been reported that the motor enzyme translocates nucleotides at a fairly consistent rate of 70-to-120 nucleotides per second. Observing 372 nucleotides after 3.5 seconds of sequencing is within the expectation given what is known about ONT's motor protein. Do the authors have a possible explanation for this deviation from the expected number of nucleotides sequenced after 1 second of sequencing?

Furthermore, it is unclear if the time-to-decision starts from the moment the molecule enters the pore (ONT adapter) or from when the live basecalling has sufficient signal to begin basecalling?

Nine-to-24 k reads is a wide range for that amount of throughput. Can the authors make the table in figure 1c clear about the total number of reads sequenced, the number of GAPDH reads, and the

number of initial nanopores in the flowcell?

All the scatter plots in figure 4 would benefit from a correlation measure, such as a sample correlation, Pearson correlation, or Spearman correlation.

There appears to be a mismatch between the number of reported pass reads in figure 1c (13k) and Supp figure 1a (7497). Were there a large number of unaligned pass reads? Can the authors explain this difference?

Furthermore, I find the bar chart in Supplemental figure 1a helpful and would like to see the distributions of accept reads for each transcript at AS sampling time with the raw read numbers in addition to the reported percentage of the total throughput.

Additionally, the length kde's for each of the IVT transcripts in Supplemental figure 1b-e are informative, and length distributions for each of the AS sampling times would be helpful to include in those plots as well.

As the authors have stated, direct RNA nanopore sequencing proceeds in the 3' to 5' direction, but it appears that there is much higher coverage at the 5' end of the 18s rRNA in figure 1d. Did the authors include an in vitro polyadenylation step after IVT synthesis and prior to library construction? The methods section does not indicate that was done. If not, do the authors have another explanation for why there is higher coverage at the 5' end of the 18s rRNA?

In the control experiment where 50% of the pores were in bulk mode and 50% were in AS mode, can the authors report the total number of reads from each portion of the flowcell in addition to the number of reads that matched the target?

When describing the sequencing experiment with *Candida albicans*, reporting both the raw numbers and the percentages for the enrichment and false rejection would be helpful for better understanding the tools effectiveness. Also knowing how many unique transcripts were rejected in addition to the 319 targeted transcripts for enrichment would be helpful to know.

The claim in the first sentence of the discussion should have a citation "The discovery of new and poorly abundant transcripts, as well as their gene organization, remains a challenge in transcriptomics because RNA expression varies by 10⁵"

Eighteen citations are low for literature available in the field of transcript enrichment/depletion, Nanopore sequencing, direct RNA Nanopore sequencing, and Nanopore adaptive sampling. A more detailed introduction and discussion would demonstrate that the authors are more aware of the field and where their adaptive sampling efforts fit into the broader field of selective transcriptomics.

Two minor points, all Latin words, such as "in vitro" should be in italics, and throughout the manuscript, the apostrophe is used instead of the prime character to denote the 3' and 5' ends of nucleotide strands.

We thank the reviewers for their comments, which have made our manuscript better. Below are our point by point comments.

Reviewer #1 (Remarks to the Author):

In this paper Yang and colleagues test aspects of adaptive sampling on RNA sequencing on Oxford Nanopore sequencers. They use a mixture of synthetic and real samples to determine the relative benefits of the approach and how one might optimise the approach. The data within the paper are interesting and useful.

However, the paper suffers from a common flaw in adaptive sampling papers in that at no point is absolute enrichment calculated for the experiments presented. The majority of data are presented as proportions or some other normalised variant. This does not help the reader (see below for a detailed discussion of figure 1). In my view it is essential to present enrichment in absolute terms using the number of bases sequenced on target in adaptive sampling experiments compared with control experiments. As shown below this highlights variability in experiments that users may expect to see. Crucially, this removes the reliance on the proportion of reads in the accepted or rejected fraction which is really a measure of the efficiency of read classification - it is not a measure of the efficiency of adaptive sampling.

We thank the reviewer for his/her positive comments. We have now resequenced all of the *in vitro* transcribed RNA enrichment and depletion samples in Figures 1 and 2 using 50% of the pores for bulk sequencing and 50% of the pores for adaptive sampling in the same flow cell. This allows us to present the data in absolute terms of the number of reads that are sequenced on target in adaptive sampling compared with control experiments. The number of bases sequenced in adaptive sampling is also provided in supplementary tables 1 and 2.

Figure 1 A-C is clear.

Figure 1D is also clear although it is unclear why the coverage for 18S rRNA does not drop from 3' to 5' as every other transcript does?

We thank the reviewer for his/her comments. Interestingly, we observed that there are at least 4 regions with multiple As within the 18S rRNA sequence (see **Figure 1 below**). We suspect that in addition to the final polyA sequence, the Nanopore sequencing adaptor can also bind to internal As to initiate sequencing. To test this, we reverse transcribed 18S rRNA using a reverse primer that contains 15Ts joined to a nanopore sequencing adapter sequence. We then performed PCR amplification using

Figure 1. Internal priming by oligoDT could result in nanopore sequencing starting within the 18S rRNA sequence. 18S rRNA that contains 20As was reverse transcribed using a reverse primer that contains an oligo-dT sequence that is joined with a nanopore adapter sequence. We then performed PCR using 18S rRNA specific forward primer and nanopore adapter specific reverse primer, and ran the products out on an agarose gel. We observed multiple bands that are smaller in size than the full length 18S rRNA, likely due to internal priming of oligoDT because of internal As.

18S rRNA specific forward primer and the nanopore adapter reverse primer and ran the PCR products onto an agarose gel. We observed several major bands corresponding to the full length 18S rRNA product, as well as to the shorter PCR products (1kb, 700 bases etc) that are likely due to internal priming.

Figure 1E and F are somewhat confusing as the data are presented grouped by transcript (E) or grouped by time (F). Why is this the case? The data are difficult to intuitively understand. The authors propose a 2.3 fold increase in the “amount of GAPDH that is present in the accepted pool”. This statement is misleading. The accepted pool is defined as those reads with a stop receiving classification or a no decision classification. If every read was correctly classified, then all GAPDH reads would be in the accepted pool. This would give a proportion of 100% reads in the accepted pool but tells us nothing about an increase in amount. In order to know this, one needs to know read numbers and numbers of bases. Read numbers are reported in supplementary table 1. This does indeed show that some absolute enrichment is seen in the number of reads that are observed in two of the experiments - in the 1 second AS time, there are 1922 accepted GAPDH reads (i.e a 1.7 fold increase). In the 3.5s AS experiment there are 2187 reads (i.e a 1.96 fold improvement. However, in the 6.5s experiment there are only 791 reads - i.e a drop in yield (a 0.71 fold “improvement”).

We thank the reviewer for his/her comments. We agree with the reviewer that enrichment in the accepted reads pool can be misleading. As such, we have now resequenced the *in vitro* transcribed samples, using 50% of the pores for bulk sequencing and 50% of the pores for adaptive sequencing, at 1, 3.5 and 6.5 sec of decision times (see **Figure 2 below** or see **Figure 1e, f, g** in the manuscript). At each decision time, we calculated the absolute number of reads that map onto the GAPDH transcript in bulk sequencing versus in adaptive sampling. The number of bases sequenced in adaptive sampling is also provided in supplementary tables 1 and 2. GAPDH originally takes up ~16.5% of the reads in the pool. By enriching for GAPDH, we observed an increase in the absolute number of reads mapping to GAPDH at 3.5 sec of decision time (16227 reads in bulk sequencing versus 19867 reads in adaptive

Figure 2. Number of reads obtained on ENO2, 18S rRNA, Actin B and GAPDH with and without enriching for GAPDH, using 1s (Top), 3.5s (middle) and 6.5s (bottom) as decision times. The right panels indicate bar plots that show the proportion of reads on each of the 4 genes upon adaptive sampling.

sampling), indicating a 22% increase in read count. We did not observe an increase in reads on GAPDH at 1 sec of decision time probably because 1 sec is insufficient to deplete non-target transcripts effectively. Additionally, we observed a smaller increase in an absolute number of reads to GAPDH at 6.5 sec decision time, probably 6.5 sec of sequencing results in too many bases being sequenced before decision time, for adaptive sampling to be effective.

This most likely reflects the yield of each of the experiments. Indeed - the control experiment generates a total of 9,225 reads, the 1s 20,363 (including rejected reads), the 3.5s 20,086 and the 6.5s only 7,745 reads. Thus the drop in yield is the underlying explanation for the different result. Remarkably if one calculates the relative proportions of reads mapping to each transcript in the pools and includes sequenced and rejected counts, one finds that the relative proportions of reads identified as coming from each transcript doesn't significantly change (exactly as one expects!). It is merely the length of the reads that will differ.

We thank the reviewer for his/her comments. We have now resequenced the samples to make sure that a similar and adequate number of reads were obtained at each time point (see **Figure 3** below or see **Figure 1c, Supp. Figure 1a, b** in the manuscript). This allows us to compare the bulk versus adaptive sampling results fairly across different decision times.

Figure 3. a, No. of reads sequenced of GAPDH without adaptive sampling and at different decision times of adaptive sampling. **b**, Bar plots showing the average number of reads of GAPDH, ENO2, 18S rRNA, and ACTB present in bulk sequencing. **c**, Bar plots showing the reads count from bulk and 1sec, 3.5sec and 6.5sec adaptive sequencing.

In my view the experiment has done exactly what it should have done, but that is hard to see in the way it is presented. The measurements of “enrichment” are not accurate for either read number or bases sequenced as presented.

We thank the reviewer for his/her positive comments. We have now resequenced the entire IVT pool experiment and displayed the data as absolute read counts. The number of bases sequenced in adaptive sampling is also provided in supplementary tables 1 and 2.

The same presentation arguments apply to figure 2. Indeed, I think the same issues are present throughout the remainder of the manuscript. By measuring the fraction of reads in the accepted pool the analysis fails to fairly represent the total data.

We thank the reviewer for his/her comments. We have now also repeated the experiments to deplete ENO2 using 50% adaptive sampling and 50% bulk sequencing to calculate the absolute number of reads on the IVT RNAs. The number of bases sequenced in adaptive sampling is also provided in supplementary table 2.

We observe that 2 sec of decision time is insufficient to deplete ENO2 because of the longer transcripts of ENO2 still being sequenced (see **Figure 4 below** or see **Figure 2a in the manuscript**). Between 2.5, 3.5 and 4.5 sec of decision time, we observed that 3.5 sec is sufficient to decrease the number of ENO2 sequenced reads to 21% of that in bulk sequencing (see **Figure 5 below** or see **Figure 2c-e in the manuscript**). The absolute number of reads of the other 3 transcripts in the population increased from 11865 to 15311 (18S rRNA), 3656 to 4595 (ACTB) and 13323 to 17268 (GAPDH) upon depletion of ENO2. This represents an absolute increase of 26-30% in the number of reads on the other three transcripts.

Figure 4. No. of reads along the length of the ENO2 transcript with bulk sequencing and with adaptive sampling at 2, 2.5, 3.5 and 4.5 sec decision times to deplete it

Figure 5. Number of reads obtained on ENO2, 18S rRNA, Actin B and GAPDH with and without depleting for ENO2, using 2.5s (Top), 3.5s (middle) and 4.5s (bottom) as decision times. The right panels indicate bar plots that show the proportion of reads on each of the 4 genes upon adaptive sampling.

In figure 3 panels D, H and I the absolute change in read number is relatively small. Is there a way to determine the significance of this? In the absence of measures of absolute yield it is difficult to determine if the read number measures are sufficient.

We thank the reviewer for his/her comments. To determine whether the absolute change in read number upon adaptive sampling is significant, we performed the Wilcoxon Rank Sum Test on the number of reads on the on-target genes with and without adaptive sampling (see **Figure 6** below or **Supp. Figure 3c, d, e**). We observed that enriching 319 genes did not result in a significant difference in the number of reads on the 319 genes ($p=0.06$), while depleting 95% of the genes did significantly

Figure 6. Boxplots showing the number of reads on on-target genes in bulk sequencing and in adaptive sampling. We observed a significant increase in the number of reads on the 319 genes upon depletion of 95% of the transcriptome (middle). We also observed a significant increase in the number of reads on the rest of the genes upon depleting 150 top genes (right).

increase the number of reads on 319 genes ($p=0.00032$). Additionally, depleting 150 genes also resulted in a significant increase in the number of reads on the rest of the transcripts ($p=0.002$).

Reviewer #2 (Remarks to the Author):

The authors of “Direct RNA sequencing coupled with adaptive sampling enriches RNAs of interest in the transcriptome” demonstrate that adaptive sampling applied to direct RNA nanopore sequencing can enrich or deplete target transcripts during sequencing without the need for biochemical techniques prior to sequencing. They used both a sample of 4 different IVT transcripts of similar lengths and the fungus *Candida albicans*. They demonstrated that the adaptive sampling of RNA molecules doesn’t unnecessarily degrade the nanopore and that there are significant fold enrichments of target transcripts when compared to bulk phase. However, as stated in the discussion by the authors, the time to decision is too long and at best results in only a 10% increase in number of on target reads, suggesting that another flowcell worth of sequencing would result in more on and off target transcripts sequenced than if adaptive sampling was used. There needs to be extensive improvements in the time to decision before adaptive sampling can effectively be used with direct RNA nanopore sequencing. This will be of even more importance if ONT begins using their new faster RNA motor as advertised last summer.

We thank the reviewer for his/her insightful comments.

Below are specific comments to the authors:

The introduction is sparse on details that would prepare the reader for the experiments to follow. More details on the effectiveness of the various enrichment or depletion methods mentioned in the introduction would help put in perspective what is possible and where this method fits into the field of target transcript enrichment/depletion and DNA adaptive sampling.

We thank the reviewer for his/her comments. We have now expanded our introduction.

The claim in the second sentence of the introduction should have one or more citations. “However, the most abundant 100 transcripts typically take up more than 50% of the sequencing reads, making it difficult to detect specific transcripts of interest such as long non coding RNAs, and to discover new transcripts that were previously undetectable.”

We thank the reviewer for his/her comments. We have now included citations on this.

The claim in the last sentence of the first paragraph of the introduction should have one or more citations “While effective in certain cases, these methods can suffer from poor enrichment efficiencies and degradation of RNA material during the enrichment steps, complicating the experimental process.”

We thank the reviewer for his/her comments. We have now included citations on this.

It is unclear from the initial testing with the 4 IVT RNAs (ACTB, GAPDH, 18s rRNA, and ENO2) if the poly(A) tail was included as part of the decision time, and how different poly(A) tail lengths may influence the method’s enrichment or depletion of targets.

We thank the reviewer for his/her concerns. ACTB, GAPDH and 18S rRNA each have 20As at the end of the RNA, and ENO2 (which is included as a positive control in direct RNA sequencing) has 30As at the end.

PolyA tail length is included as part of the decision time. To determine whether different polyA tail lengths may influence the method’s enrichment or depletion of targets, we used nanopolish to estimate the length of polyA tails in the *Candida albicans* transcriptome. We then sorted transcripts according to their polyA tail lengths. As the overall lengths of polyA tails in *Candida albicans* is relatively short (<70 bases), we did not observe a significant relationship between polyA length and adaptive sequencing (see **Figure 7** below or **Supp. Figure 4c** in the manuscript).

Figure 7. Scatterplot showing the correlation between polyA tail length and the ability of the transcript to be rejected efficiently. We did not observe a significant contribution of polyA tail length to the ability of the transcript to be rejected.

I’m a little surprised that the median number of nucleotides sequenced for the rejected transcripts were 276 for 1 second of sequencing. It has been reported that the motor enzyme translocates nucleotides at a fairly consistent rate of 70-to-120 nucleotides per second. Observing 372 nucleotides after 3.5 seconds of sequencing is within the expectation given what is known about ONT’s motor protein. Do the authors have a possible explanation for this deviation from the expected number of nucleotides sequenced after 1 second of sequencing?

We thank the reviewer for his/her comments.

The decision time parameter (from 1s to 6.5s) controls the polling time of the sequenced read. At each stated decision time interval, the traces are presented for basecalling and alignment so that a decision can be made. By the time a decision is made, the time taken is the sum of 1 sec of sequencing plus the time needed for basecalling and alignment. Here, the median result of 276-nt does not refer to only 1s of sequencing, but to the final read length after a decision has been made. If a decision cannot be made, the pore continues sequencing where it left off for another decision time interval.

Furthermore, it is unclear if the time-to-decision starts from the moment the molecule enters the pore (ONT adapter) or from when the live basecalling has sufficient signal to begin basecalling?

Time to decision starts from the moment the strand enters the pore and does include time necessary to traverse the adapter sequence.

Nine-to-24 k reads is a wide range for that amount of throughput. Can the authors make the table in figure 1c clear about the total number of reads sequenced, the number of GAPDH reads, and the number of initial nanopores in the flowcell?

We thank the reviewer for his/her comments. We have now resequenced all of our IVT experiments, using 50% adaptive sampling and 50% bulk sequencing on the same flow cell, so that we control for variations in the number of active pores in different flow cells. We obtained a similar number of reads across different conditions (see **Figure 1c, Supp. Figure 1a, b, and Supplementary Table 1** in the manuscript).

At 1 sec decision time, we sequenced a total of 92467 mapped reads from bulk sequencing, out of which 15273 reads to GAPDH. Using adaptive sampling, we obtained 12197 accepted reads for GAPDH, suggesting that adaptive sampling does not increase the absolute number of GAPDH reads at 1 sec decision time.

At 3.5 sec decision time, we sequenced a total of 103259 reads for bulk sequencing, out of which 16227 reads mapped to GAPDH. Using adaptive sampling, we obtained 19867 accepted reads for GAPDH, indicating a 22% increase in reads.

At 6.5 sec decision time, we sequenced a total of 97135 reads for bulk sequencing, out of which 16149 reads mapped to GAPDH. Using adaptive sampling, we obtained 17252 reads mapping to GAPDH, indicating a 7% increase in reads. 6.5 sec is too long because too many bases would have been sequenced before a decision is made to reject or accept a read, making adaptive sampling less effective.

We have also resequenced the IVT experiment using 50% bulk sequencing and 50% adaptive sampling for the depletion mode of adaptive sampling in **Figure 2 and Supp. Table 2** in the manuscript. On average we sequenced ~80K reads per decision time point (82076, 88790, 85539, 76681 bulk sequencing reads for 2, 2.5, 3.5 and 4.5 sec decision times respectively).

All the scatter plots in figure 4 would benefit from a correlation measure, such as a sample correlation, Pearson correlation, or Spearman correlation.

We have now added correlations to both **Figures 3 and 4** in the manuscript.

There appears to be a mismatch between the number of reported pass reads in figure 1c (13k) and Supp figure 1a (7497). Were there a large number of unaligned pass reads? Can the authors explain this difference?

We thank the reviewer for pointing this out. We accidentally reported the number from an older version of sequencing run and have now updated this in the manuscript. We do not see a large number of unaligned pass reads (**Figure 8** below or **Figure 1c** and **Supp. Figure 1a** in the manuscript).

TARGET	GAPDH	GAPDH	GAPDH
BREAK TIME(S)	1 (default)	3.5	6.5
total Aligned reads from AS	195K	244K	211K
total Aligned reads from Bulk	92.5K	103K	97K
GAPDH reads from AS	12.2K	19.9K	17.3K
GAPDH reads from bulk	15.3K	16.2K	16.1K

Figure 8. *Top*, table showing the number of total reads counts and the reads mapped onto GAPDH in bulk sequencing and in adaptive sampling, at different decision times. *Bottom*, bar plot showing the number of reads mapped to each of the 4 RNAs in bulk sequencing.

Furthermore, I find the bar chart in Supplemental figure 1a helpful and would like to see the distributions of accept reads for each transcript at AS sampling time with the raw read numbers in addition to the reported percentage of the total throughput.

We thank the reviewer for his/her comments. As mentioned, we have now resequenced our IVT library using 50/50 bulk and adaptive RNA sampling to enable direct comparisons of the number of reads mapped to GAPDH in adaptive sampling versus bulk sequencing mode.

We have now plotted the number of reads that are mapped to each RNA in bulk sequencing and adaptive sampling (at different decision times) upon enriching for GAPDH (see **Figure 9a-c** below or **Figure 1e, f, g** in the manuscript).

Figure 9. Barplots showing the number of reads on each of the 4 RNAs during bulk sequencing and during adaptive sampling, at 1sec (a), 3.5sec (b) and 6.5sec (c) of decision times to enrich for GAPDH.

Addi are informative, and length distributions for each of the AS sampling times would be helpful to include in those plots as well.

We have now plotted the density plots of their length distributions in bulk sequencing and in adaptive sampling (see **Figure 10** below or **Supp. Figure 1c-f** in the manuscript).

Figure 10. Density plot showing the length distributions of the four RNAs in bulk sequencing and upon enrichment of GAPDH.

As the authors have stated, direct RNA nanopore sequencing proceeds in the 3' to 5' direction, but it appears that there is much higher coverage at the 5' end of the 18s rRNA in figure 1d. Did the authors include an in vitro polyadenylation step after IVT synthesis and prior to library construction? The methods section does not indicate that was done. If not, do the authors have another explanation for why there is higher coverage at the 5' end of the 18s rRNA?

We thank the reviewer for his/her comments. Interestingly, we observed that there are at least 4 regions with multiple As within the 18S rRNA sequence (see **Figure 11** below). We suspect that in addition to the final polyA sequence, the Nanopore sequencing adaptor can also bind to internal As to initiate sequencing. To test this, we reverse transcribed 18S rRNA using a reverse primer that contains 15Ts joint to a nanopore sequencing adaptor sequence. We then performed PCR amplification using 18S rRNA specific forward primer and the nanopore adapter reverse primer and ran the PCR products onto an agarose gel. We observed several major bands corresponding to the full-length 18S rRNA product, as well as shorter PCR products (1kb, 700 bases, etc) that are likely due to internal priming.

Figure 11. Internal priming by oligoDT could result in nanopore sequencing starting within the 18S rRNA sequence. 18S rRNA that contains 20As was reverse transcribed using a reverse primer that contains an oligo-dT sequence that is joined with a nanopore adaptor sequence. We then performed PCR using 18S rRNA specific forward primer and nanopore adapter specific reverse primer, and ran the products out on an agarose gel. We observed multiple bands that are smaller in size than the full length 18S rRNA, likely due to internal priming of oligoDT because of internal As.

In the control experiment where 50% of the pores were in bulk mode and 50% were in AS mode, can the authors report the total number of reads from each portion of the flowcell in addition to the number of reads that matched the target?

We have included all of the data from the absolute reads from 50% of the pores in bulk mode and 50% in adaptive sampling mode as Supplementary Tables (see **Supp. Tables 1, 2**).

When describing the sequencing experiment with *Candida albicans*, reporting both the raw numbers and the percentages for the enrichment and false rejection would be helpful for better understanding the tools effectiveness. Also knowing how many unique transcripts were rejected in addition to the 319 targeted transcripts for enrichment would be helpful to know.

We have included all of the data from the absolute reads from 50% of the pores in bulk mode and 50% in adaptive sampling mode as Supplementary Tables (Supp. Tables 3, 4, 5). When we enrich for 319 transcripts, 220357 reads were rejected, including 4112 reads that belong to the 319 genes (1.8% of all rejected reads). When we deplete 95% of the transcriptome to enrich for the 319 genes, 419130 reads are rejected, of which 1355 belong to 319 genes. We have shown this as Figure 3c, e in the manuscript.

The claim in the first sentence of the discussion should have a citation “The discovery of new and poorly abundant transcripts, as well as their gene organization, remains a challenge in transcriptomics because RNA expression varies by 10^5 ”

We have now included citations for this claim.

Eighteen citations are low for literature available in the field of transcript enrichment/depletion, Nanopore sequencing, direct RNA Nanopore sequencing, and Nanopore adaptive sampling. A more detailed introduction and discussion would demonstrate that the authors are more aware of the field and where their adaptive sampling efforts fit into the broader field of selective transcriptomics.

We have now expanded the introduction and the number of citations for this manuscript.

Two minor points, all Latin words, such as “*in vitro*” should be in italics, and throughout the manuscript, the apostrophe is used instead of the prime character to denote the 3' and 5' ends of nucleotide strands.

We have now corrected this in the manuscript.

REVIEWER COMMENTS

Reviewer #1 (Remarks to the Author):

The manuscript from Wang and colleagues has made several amendments in line with previous comments I made on the manuscript.

However there are still critical flaws in the analysis as it is presented. Whilst the authors state they agree with my assessment that the "accepted read pool" metric is misleading, they continue to use it in the manuscript. The section entitled "Adaptive sampling enriches RNA of interest in the accepted read population." should be renamed. What the investigators are measuring here is how well they can identify reads from a given length of signal. This section is confused as it is conflating two different measurements. The experiment where they divide the flow cell into two halves does investigate the consequences of adaptive sampling on enrichment (finding an enrichment of 1.22x).

The authors then go on to use the same accepted pool metric in the section on depletion. Again this is confusing and misleading. By definition the depletion mode as described by the authors requires that reads be mapped to the target and then removed. Thus these reads must be long enough to be called and mapped to the sample. If so, then they should be called and mapped in the final output? Indeed - I think the reads are in the final output - they are simply being binned into the "rejected pool". But this is simply a classification issue.

As in my previous comments, the authors are still not clearly showing the total amount of data generated in any experiment. Read counts are meaningless in a system where reads can be of different lengths. It is yield (i.e number of bases) on or off target that is the only valid metric.

The authors move on from the model system they are looking at (four transcripts) to consider *Candida albicans*. Here they show that the top 150 transcripts take up 55% of the sample (in contrast to the statement they make in sentence 2 of the introduction). The authors again reference the accepted read pool even though they acknowledge this is a misleading measure. Indeed, in a correctly controlled experiment, the authors show that this experiment does not lead to enrichment (indeed - the selection for 319 transcripts results in FEWER reads on target - i.e enrichment has resulted in depletion. In the inverse experiment (depleting the other targets) the authors do show a small increase in the number of reads seen (13.4%). It is not clear here which metric the authors are showing - is it the proportion of reads in the whole data set or that in the accepted or rejected read pool?

The authors then carry out a depletion experiment on the 150 most abundant targets - the observations are yet again confounded by breaking the data into the accepted and rejected read pools, but the authors do show an increase of 11.5% in reads from the rest of the transcriptome.

The final part of the manuscript investigates the detection of 17 novel transcripts as a result of depleting all known transcripts. The experiment is minimally described with no apparent controls. How many additional reads were sequenced? What was the yield and throughput in the experiments? How many of these transcripts could be found just by sequencing more reads?

In the abstract the authors claim an increase of 2.3-3.2 fold in the proportion of reads in a library as a result of adaptive sampling. This is nonsensical. The proportion of reads mapping to something cannot change without chemical manipulation. It is simply impossible and is a misleading claim. As discussed above, the metric of the "accepted read pool" is simply a measure of classification accuracy and does not reflect a true enrichment. The 13.5% increase in read number is potentially advantageous, but is it worth the effort?

The discovery of new transcripts is potentially of interest, but this is under described in the manuscript.

The authors appear to accept the use of the arbitrary "accepted read pool" is misleading, but they continue to use it. This can only be used as a measure of the efficacy of reads being identified by adaptive sampling. If the authors wish to measure enrichment or depletion they must use a metric which fairly reflects the metrics - I would suggest measures of mean read length and bases on target. In addition, the total yield (in bases, not reads) from each experimental condition (i.e flow cell or 50% of flowcell etc) regardless of reads being accepted or rejected should be stated.

In simple terms, does a user gain more insight if they use this method or run another flow cell?

Reviewer #2 (Remarks to the Author):

The authors of "Direct RNA sequencing coupled with adaptive sampling enriches RNAs of interest in the transcriptome", have significantly improved this version of the manuscript, especially the introduction. It now properly sets up the reader for what to expect from the rest of the study, and better establishes the context for this work. The adaptive sampling appears to work better when in depletion mode, but there are still somethings that are not entirely clear to me when working in enrichment mode.

In Figure 1 the authors found that they had a 22% increase in GAPDH reads in the adaptive sampling mode (AS), compared to Bulk. But this measure is not normalized by the coverage differences in the two halves of the experiment, when looking at the number of GAPDH reads as a percentage of the total reads for the half, the percentages are roughly the same.

TARGET GAPDH GAPDH GAPDH

BREAK TIMES (S) 1 (default) percentage of (AS or Bulk) 3.5 percentage of (AS or Bulk) 6.5
percentage of (AS or Bulk)

Total aligned reads from AS 102k - 140k - 114k -

Total aligned reads from Bulk 92,5k - 103k - 97k -

GAPDH reads from AS 12,2k 11,96% 19,9k 14,21% 17,3k 15,18%

GAPDH reads from Bulk 15,3k 16,54% 16,2k 15,73% 16,1k 16,60%

*Adapted from Figure 1c

A plot like that in Supplementary Figure 5b, but for the number of bases sequenced for each of the 4 IVT transcripts might help clarify this.

The author's explanation about internal priming on the 18s rRNA does not make sense given how the Direct RNA nanopore sequencing adapters are ligated to the poly(A) tail. Because the RNA is sequenced directly, internal priming would only make sense if the adapter was being ligated to the 3' end of the rRNA at those internal poly(A) sites. Their PCR experiment partially addresses this concern but is not fully convincing that the ONT adapter is being ligated to these internal poly(A) sites. A better experiment to test this hypothesis would have been to follow the ONT direct RNA nanopore library preparation protocol up to the cDNA synthesis step. Then use the 5' primer they used in the experiment and the ONT adapter sequence as the reverse priming site. In this way, it would more accurately test the idea of internal priming for direct RNA nanopore sequencing, rather than how it would work for the cDNA. And a control for this experiment would be 18s rRNA that followed all the same steps, except that the ligase for the ONT adapter ligation step was omitted. Did the authors review the sequence at the 3' ends of their internally primed 18s rRNA direct RNA nanopore reads and see that they started near the poly(A) sites? A PAGE gel of the IVT products might help in

understanding the increased coverage at the 5' end.

A suggestion for figure 4a and 4b. I think the message from these plots are clear, but perhaps plotting the ratio of the Fraction of reads in accept pool to the fraction of reads in bulk on the y axis, and the length of the transcript on the x axis might more clearly demonstrate that the shorter transcripts are less efficiently enriched/depleted. This could also potentially collapse the 4 plots into 1.

In figure 4c and 4d, it appears that the novel transcripts also appear in the bulk sequencing. Did the authors try to find novel transcripts in bulk sequencing as well as the AS sequencing?

We thank the reviewer for his/her comments, which has made the manuscript better.

Reviewer #1 (Remarks to the Author):

The manuscript from Wang and colleagues has made several amendments in line with previous comments I made on the manuscript.

However there are still critical flaws in the analysis as it is presented. Whilst the authors state they agree with my assessment that the “accepted read pool” metric is misleading, they continue to use it in the manuscript. The section entitled “Adaptive sampling enriches RNA of interest in the accepted read population.” should be renamed. What the investigators are measuring here is how well they can identify reads from a given length of signal. This section is confused as it is conflating two different measurements. The experiment where they divide the flow cell into two halves does investigate the consequences of adaptive sampling on enrichment (finding an enrichment of 1.22x).

The authors then go on to use the same accepted pool metric in the section on depletion. Again this is confusing and misleading. By definition the depletion mode as described by the authors requires that reads be mapped to the target and then removed. Thus these reads must be long enough to be called and mapped to the sample. If so, then they should be called and mapped in the final output? Indeed - I think the reads are in the final output - they are simply being binned into the “rejected pool”. But this is simply a classification issue.

As in my previous comments, the authors are still not clearly showing the total amount of data generated in any experiment. Read counts are meaningless in a system where reads can be of different lengths. It is yield (i.e number of bases) on or off target that is the only valid metric.

We thank the reviewer for his/her comments. As suggested by the reviewer, we have now removed all data showing enrichment in the accepted pool and instead present the data as either the absolute number of reads or bases in our manuscript (see **Figure 1c, e, f, g** in the manuscript). Using 50-50% bulk sequencing and adaptive sequencing, we observed that enriching for GAPDH resulted in a 28% increase in the number of bases and 22% increase in the number of reads mapped to GAPDH in adaptive sampling (3.5 sec decision time) as compared to bulk sequencing in the IVT pool (see **Figure 1** below).

Figure 1. Schematic showing parameters for bulk and adaptive sequencing, using 50% of the pores of a flow cell for bulk sequencing and 50% of the pores for adaptive sequencing at 3.5 sec of decision time. Left Middle: Bar plots showing the total number of bases obtained on GAPDH, ENO2, 18S rRNA, and ACTB using adaptive sampling and bulk sequencing. Right Middle: Bar plots showing the number of reads mapped to GAPDH, ENO2, 18S rRNA, and ACTB in bulk sequencing and in adaptive sampling sequencing. Right: Bar plots showing the percentage of accepted reads in adaptive sequencing as compared to bulk sequencing for each transcript.

We have now included the number of bases obtained by adaptive sampling and bulk sequencing in the IVT sample in **Figures 1e-g** in the manuscript (see **Figure 2** next page).

Figure 2. Bar plots showing the total number of bases obtained on GAPDH, ENO2, 18S rRNA, and ACTB using adaptive sampling and bulk sequencing at 1sec (left), 3.5sec (middle), 6.5sec (right) of decision time.

Similarly in the depletion mode, depleting ENO2 resulted in an ~30% increase in the total number of bases and reads of the other three RNAs (18S rRNA, ACTB and GAPDH). We plotted the number of bases obtained on all four RNAs at each decision time point (see Figure 3 below). We have now included this data in our Figure 2b, c, d, e, g in the manuscript.

Figure 3. Bar plots showing the total number of bases obtained on GAPDH, ENO2, 18S rRNA, and ACTB using adaptive sampling (to deplete ENO2) and bulk sequencing at 2sec (top left), 2.5sec (top middle), 3.5sec (top right), and 4.5sec (bottom left) of decision time. We also show bar plots for the total number of bases obtained on GAPDH, ENO2, 18S rRNA, and ACTB using adaptive sampling (to deplete ENO2 and GAPDH) and bulk sequencing (bottom right).

The authors move on from the model system they are looking at (four transcripts) to consider *Candida albicans*. Here they show that the top 150 transcripts take up 55% of the sample (in contrast to the statement they make in sentence 2 of the introduction). The authors again reference the accepted read pool even though they acknowledge this is a misleading measure. Indeed, in a correctly controlled experiment, the authors show that this experiment does not lead to enrichment (indeed - the selection for 319 transcripts results in FEWER reads on target - i.e enrichment has resulted in depletion. In the inverse experiment (depleting the other targets) the authors do show a small increase in the number of reads seen (13.4%). It is not clear here which metric the authors are showing - is it the proportion of reads in the whole data set or that in the accepted or rejected read pool?

We thank the reviewer for his/her comments. In the manuscript, we showed that depleting the rest of the transcriptome resulted an absolute increase in 13.4% of the reads mapped to the 319 genes as compared to bulk sequencing. We had extracted the reads from the accepted read pool. To examine the total number of reads mapped to 319 genes in adaptive sampling (regardless of whether they are

in accepted read pool or not), we summed the number of reads mapped to both the accepted and rejected pool. In addition to the 35057 reads in the accepted pool, we also observed 1355 reads in the rejected pool, this resulted in a total of 36412 reads in adaptive sampling, as compared to 30912 reads in bulk sequencing (17.8% increase, see **Figure 4** below).

Figure 4. Bar plots showing the number of bases obtained through bulk sequencing or adaptive sampling on 319 genes after depleting 95% of the transcriptome.

The authors then carry out a depletion experiment on the 150 most abundant targets - the observations are yet again confounded by breaking the data into the accepted and rejected read pools, but the authors do show an increase of 11.5% in reads from the rest of the transcriptome.

As above, depleting 150 most abundant transcripts resulted in 415610 reads on the remaining genes in the accepted pool, as compared to 373940 reads on these genes in bulk sequencing. This resulted in a 11.1% increase in the number of reads in adaptive sampling. If we include the reads on the remaining genes in both the accepted and rejected pool, this results in an increase of 4088 reads (originally from the rejected pool) and a total of 419698 reads in adaptive sampling (see **Figure 5** below). This is a 12.2% increase in adaptive sampling as compared to bulk sequencing.

Figure 5. Bar plots showing the number of reads mapped on to the rest of the transcriptome when the top 150 genes are depleted.

The final part of the manuscript investigates the detection of 17 novel transcripts as a result of depleting all known transcripts. The experiment is minimally described with no apparent controls. How many additional reads were sequenced? What was the yield and throughput in the experiments? How many of these transcripts could be found just by sequencing more reads?

We thank the reviewer for his/her comments. After checking for these transcripts in bulk sequencing, we observed that the novel genes are present in both the bulk sequencing and adaptive sampling conditions, and we did not observe an enrichment in the number of reads on the novel transcripts under adaptive sampling conditions. To examine why we did not see an enrichment of these novel genes upon depleting the transcriptome, while we observed an enrichment for the 319 transcripts when we depleted 95% of the transcriptome, we analyzed the effect of RNA abundance with regards to its ability to be enriched during adaptive sampling. Binning transcripts according to their abundance showed that transcripts with very low read counts are less enriched upon depletion of other transcripts. As the 26 novel genes are poorly expressed, 11 of them have 0-20 reads, and another 11 of them have 20-40 reads, and only 4 of them have reads between 40-60 reads, we believe that the abundance plays

a role in their lack of enrichment in adaptive sampling. Additionally, we have previously observed that longer transcripts tend to be enriched better (see **Figure 4b** in the manuscript). We also observed that our novel transcripts tend to be shorter than other known transcripts, probably further contributing to it not being enriched well during adaptive sampling (see **Figure 6** below). We have now included the data below as **Supp. Figure 4d, and 5b, c** in the manuscript, and expanded the section on novel transcripts in the manuscript to discuss this.

Figure 6. Left, Barplots showing slope of linear regression of reads count between adaptive sampling and Bulk belonging to the rest of the transcriptome, after the top 150 transcripts are depleted, binned by transcript abundance (<10, 10-20, 20-50, 50-200, 200-500, >500 reads). Middle, density plots showing the distribution of the abundance of the 26 novel transcripts as compared to annotated transcripts. Right, density plots showing the distribution of the transcript length of the 26 novel transcripts as compared to annotated transcripts.

In the abstract the authors claim an increase of 2.3-3.2 fold in the proportion of reads in a library as a result of adaptive sampling. This is nonsensical. The proportion of reads mapping to something cannot change without chemical manipulation. It is simply impossible and is a misleading claim. As discussed above, the metric of the “accepted read pool” is simply a measure of classification accuracy and does not reflect a true enrichment. The 13.5% increase in read number is potentially advantageous, but is it worth the effort?

We agree with the reviewer and have now removed these enrichments in proportion of reads from the manuscript. We have now taken a 50-50% split flow cell approach and report the absolute number of reads on the targets of interest.

The discovery of new transcripts is potentially of interest, but this is under described in the manuscript.

We thank the reviewer for his/her comments. We have now looked into the length and abundance of the new transcripts. Additionally, we also noticed that 17 out of 26 transcripts are antisense to known genes of interest, suggesting that they may play regulatory roles to these transcripts. Future experiments will need to validate the functions of these genes, which is out of the scope of this manuscript.

The authors appear to accept the use of the arbitrary “accepted read pool” is misleading, but they continue to use it. This can only be used as a measure of the efficacy of reads being identified by adaptive sampling. If the authors wish to measure enrichment or depletion they must use a metric which fairly reflects the metrics - I would suggest measures of mean read length and bases on target. In addition, the total yield (in bases, not reads) from each experimental condition (i.e flow cell or 50% of flowcell etc) regardless of reads being accepted or rejected should be stated.

We thank the reviewer for his/her comments. We have now corrected this in our manuscript.

In simple terms, does a user gain more insight if they use this method or run another flow cell?

Adaptive sampling is a way to enrich for transcripts of interest, without needing experimental biochemical enrichments before sequencing. However, like all enrichments, the transcripts of interest can always be detected with deeper sequencing. In this manuscript, we identified parameters for successful enrichments using adaptive sampling, including decision times, transcript length and abundance and observed a 20-30% increase in sequencing for transcripts of interest in an IVT pool and ~11-14% increase for transcripts in the transcriptome. While this is the first proof of concept for adaptive sampling at the RNA level, we believe that further improvements in the speed of decision making, compute time and longer lived pores will further enhance the efficiency of adaptive sampling for RNA.

Reviewer #2 (Remarks to the Author):

The authors of “Direct RNA sequencing coupled with adaptive sampling enriches RNAs of interest in the transcriptome”, have significantly improved this version of the manuscript, especially the introduction. It now properly sets up the reader for what to expect from the rest of the study, and better establishes the context for this work. The adaptive sampling appears to work better when in depletion mode, but there are still somethings that are not entirely clear to me when working in enrichment mode.

We thank the reviewer for his/her positive comments.

In Figure 1 the authors found that they had a 22% increase in GAPDH reads in the adaptive sampling mode (AS), compared to Bulk. But this measure is not normalized by the coverage differences in the two halves of the experiment, when looking at the number of GAPDH reads as a percentage of the total reads for the half, the percentages are roughly the same.

TARGET GAPDH GAPDH GAPDH

BREAK TIMES (S) 1 (default) percentage of (AS or Bulk) 3.5 percentage of (AS or Bulk) 6.5 percentage of (AS or Bulk)

Total aligned reads from AS 102k - 140k - 114k –

Total aligned reads from Bulk 92,5k - 103k - 97k –

GAPDH reads from AS 12,2k 11,96% 19,9k 14,21% 17,3k 15,18%

GAPDH reads from Bulk 15,3k 16,54% 16,2k 15,73% 16,1k 16,60%

*Adapted from Figure 1c

A plot like that in Supplementary Figure 5b, but for the number of bases sequenced for each of the 4 IVT transcripts might help clarify this.

We thank the reviewer for his/her comments. We agree that the total output of reads from the 50% of the pores performing bulk sequencing is likely to be different from the total output of reads from the 50% of pores performing adaptive sampling. From the perspective of the technology user- the number of reads that are mapped to on-target transcripts is usually the most important. As such, we have described the absolute number of on-target reads, rather than enrichment of proportions. Following the reviewer’s advice, we have now included the number of reads for each RNA in the IVT

pool in bulk adaptive sequencing (see **Figure 7** below). We have now included this in **Figure 1c** in our manuscript.

Bases count (10⁶)

	GAPDH	ACTB	18S	ENO2
1S_AS	10.1	2.2	11.7	26.1
bulk 1	11.6	4.1	15.1	43.1
3.5S_AS	17.8	2.3	7.8	34.8
bulk 2	14.1	4.3	17.0	67.6
6.5S_AS	13.1	2.3	6.6	24.1
bulk 3	12.1	4.1	15.8	43.9

Figure 7. Table showing the number of reads (in millions) mapped to each of the four RNAs in the IVT pool in bulk sequencing and adaptive sampling with different decision times.

The author's explanation about internal priming on the 18s rRNA does not make sense given how the Direct RNA nanopore sequencing adapters are ligated to the poly(A) tail. Because the RNA is sequenced directly, internal priming would only make sense if the adapter was being ligated to the 3' end of the rRNA at those internal poly(A) sites. Their PCR experiment partially addresses this concern but is not fully convincing that the ONT adapter is being ligated to these internal poly(A) sites. A better experiment to test this hypothesis would have been to follow the ONT direct RNA nanopore library preparation protocol up to the cDNA synthesis step. Then use the 5' primer they used in the experiment and the ONT adapter sequence as the reverse priming site. In this way, it would more accurately test the idea of internal priming for direct RNA nanopore sequencing, rather than how it would work for the cDNA. And a control for this experiment would be 18s rRNA that followed all the same steps, except that the ligase for the ONT adapter ligation step was omitted. Did the authors review the sequence at the 3' ends of their internally primed 18s rRNA direct RNA nanopore reads and see that they started near the poly(A) sites? A PAGE gel of the IVT products might help in understanding the increased coverage at the 5' end.

We thank the reviewer for his/her insightful comments. To determine whether we already observe truncated RNA products in our 18S rRNA during in vitro transcription, we ran the IVT products using the bioanalyzer. Indeed, we observe peaks along the bioanalyzer that corresponds to pileups in nanopore sequencing, suggesting that the truncated products could have been sequenced from the 3'ends. We then examined the ends of these read pileups in nanopore sequencing using signal alignments and observed a polyA tail, followed by the adapter sequence behind most of the pileups. As our 18S rRNA construct is first generated through reverse transcription, below performing IVT of the RNA (see **Figure 8, 9** next page), we suspect that the cloning process accidentally introduced these truncations in the cDNA constructs, resulting in these pileups in IVT and sequencing.

Figure 8. Top: Read coverage of 18S rRNA in bulk sequencing and adaptive sampling. The pile-ups in read counts are highlighted in red boxes. Middle: Bioanalyzer trace of 18S rRNA after IVT from 18S cDNA construct. The red boxes correlate to peaks in the bioanalyzer trace, indicating that the pileups we see in sequencing are probably due to shorter IVT products. Bottom: Heatmap showing signals coming from the sequenced pile-up regions. The pileup regions are mostly (>97.4%) followed by a polyA stretch and adapter sequence.

Figure 9. Expanded view of signal alignments at the end of a region of pileup. We see a stretch of polyA sequence, followed by DNA adapter sequence which shows very different signal from RNA sequence.

A suggestion for figure 4a and 4b. I think the message from these plots are clear, but perhaps plotting the ratio of the Fraction of reads in accept pool to the fraction of reads in bulk on the y axis, and the length of the transcript on the x axis might more clearly demonstrate that the shorter transcripts are less efficiently enriched/depleted. This could also potentially collapse the 4 plots into 1.

We thank the reviewer for his/her insightful comments. We have now collapsed the 4 plots for **Figure 4a** and for **Figure 4b** respectively into 2 separate plots (see **Figure 10** next page) and have now included them as our **Supp. Figure 4 a, c** in the manuscript.

Figure 10. Bar plots showing the slope of linear regression in adaptive sampling versus bulk sequencing for 95% of the transcripts being depleted (left) and for 319 transcripts that are being enriched (right). Longer transcripts tend to be depleted and enriched better.

In figure 4c and 4d, it appears that the novel transcripts also appear in the bulk sequencing. Did the authors try to find novel transcripts in bulk sequencing as well as the AS sequencing?

We thank the reviewer for his/her comments. After checking for these transcripts in bulk sequencing, we observed that the novel genes are present in both the bulk sequencing and adaptive sampling conditions, and we did not observe an enrichment in the number of reads on the novel transcripts under adaptive sampling conditions. To examine why we did not see an enrichment of these novel genes upon depleting the transcriptome, while we observed an enrichment for the 319 transcripts when we depleted 95% of the transcriptome, we analyzed the effect of RNA abundance with regards to its ability to be enriched during adaptive sampling. Binning transcripts according to their abundance showed that transcripts with very low read counts are less enriched upon depletion of other transcripts. As the 26 novel genes are poorly expressed, 11 of them have 0-20 reads, and another 11 of them have 20-40 reads, and only 4 of them have reads between 40-60 reads, we believe that the abundance plays a role in their lack of enrichment in adaptive sampling. Additionally, we have previously observed that longer transcripts tend to be enriched better (see **Figure 4b in the manuscript**). We also observed that our novel transcripts tend to be shorter than other known transcripts, probably further contributing to it not being enriched well during adaptive sampling (see **Figure 11** below). We have now included the data below as **Supp. Figure 4d**, and **5b, c**, and expanded the section on novel transcripts in the manuscript to discuss this.

Figure 11. Left, Barplots showing the ratio of reads in adaptive sampling versus bulk sequencing for transcripts with differential read coverage, after depleting the top 150 genes. Middle, density plots showing the distribution of the abundance of the 26 novel transcripts as compared to annotated transcripts. Right, density plots showing the distribution of the transcript length of the 26 novel transcripts as compared to annotated transcripts.

REVIEWERS' COMMENTS

Reviewer #1 (Remarks to the Author):

The authors have largely addressed all my comments.

However the text has not been updated to reflect the consequences of this. For example, the abstract states:

"Depleting all currently annotated *Candida albicans* transcripts revealed 26 new transcripts and isoforms, 17 of which are antisense to existing transcripts."

Yet the authors acknowledge that these same transcripts are present in the control data and - in fact - are not enriched at all. Really this experiment just finds some RNA molecules that are not previously annotated. This is a benefit of direct RNA sequencing but does not require adaptive sampling.

The sentence should probably read something like:

"Although depleting all currently annotated *Candida albicans* transcripts did not result in any absolute enrichment, we do identify 26 new transcripts and isoforms, 17 of which are antisense to existing transcripts."

In addition one can't but help think that running cDNA would be a far better approach if the goal was transcript discovery? Really a paper of this nature should probably have that as a control?

Reviewer #2 (Remarks to the Author):

Jiaxu Wang et al have made major improvements to this version of the manuscript and tempered some of the claims to be more in line with the evidence presented in the study. I agree with the authors that this study demonstrates that the infrastructure for adaptive sampling can be applied to direct RNA nanopore sequencing and has some utility, but improvements in time-to-decision are required before it should be widely adopted. The authors have addressed all my previous questions and have not introduced new questions in this draft of the manuscript. I would like to thank the authors for addressing my previous questions, especially my fixation on the internal sequencing start sites for the IVT 18s rRNA.

Minor point, I think the authors meant to say "threading" instead of "treading" in this sentence in the third paragraph in the introduction?

"Due to the slower motor speeds in ****treading**** RNA through the pores in direct RNA sequencing as compared to DNA sequencing..."

We thank the reviewers for his/her comments, which have made the manuscript better. Below are our point-by-point comments.

REVIEWERS' COMMENTS

Reviewer #1 (Remarks to the Author):

The authors have largely addressed all my comments.

We thank the reviewer for his/her positive comments.

However the text has not been updated to reflect the consequences of this. For example, the abstract states:

“Depleting all currently annotated *Candida albicans* transcripts revealed 26 new transcripts and isoforms, 17 of which are antisense to existing transcripts.”

Yet the authors acknowledge that these same transcripts are present in the control data and - in fact - are not enriched at all. Really this experiment just finds some RNA molecules that are not previously annotated. This is a benefit of direct RNA sequencing but does not require adaptive sampling.

The sentence should probably read something like:

“Although depleting all currently annotated *Candida albicans* transcripts did not result in any absolute enrichment, we do identify 26 new transcripts and isoforms, 17 of which are antisense to existing transcripts.”

We thank the reviewer for his/her insightful comment- we have now changed our abstract accordingly.

In addition one can't but help think that running cDNA would be a far better approach if the goal was transcript discovery? Really a paper of this nature should probably have that as a control?

We thank the reviewer for his/her insightful comment- unfortunately adaptive sampling cannot be used for cDNA as the motor proteins thread the cDNA through the pores too quickly (~450 bases/sec for DNA as compared to ~70 bases/sec for RNA) for adaptive sampling to be effective.

Reviewer #2 (Remarks to the Author):

Jiaxu Wang et al have made major improvements to this version of the manuscript and tempered some of the claims to be more in line with the evidence presented in the study. I agree with the authors that this study demonstrates that the infrastructure for adaptive sampling can be applied to direct RNA nanopore sequencing and has some utility, but improvements in time-to-decision are required before it should be widely adopted. The authors have addressed all my previous questions and have not introduced new questions in this draft of the manuscript. I would like to thank the authors for addressing my previous questions, especially my fixation on the internal sequencing start sites for the IVT 18s rRNA.

We thank the reviewer for his/her positive comments.

Minor point, I think the authors meant to say “threading” instead of “treading” in this sentence in the third paragraph in the introduction?

“Due to the slower motor speeds in ****treading**** RNA through the pores in direct RNA sequencing as compared to DNA sequencing...”

We have now corrected this in the manuscript.